# Knowledge, attitude and practices among Healthcare Workers towards Pulmonary Plague Infection following an outbreak in Madagascar, 2017: A pilot study

Inessa Markus[1,2☯], Lynn Meurs[1,2☯], Rebekah Wood[3], Vaoary Razafimbia[4], Andrianasolo Radonirina[5], Delphin Kolie[6], Rapelerano Rabenja Fahafahantsoa[5], Alexandre Delamou[7], Jan Walter[1], Matthias Borchert[3‡], Thomas Paerisch[8‡*]

1 Department of Infectious Disease Epidemiology, Robert Koch Institute, Berlin, Germany, 2 ECDC Fellowship Programme, Field Epidemiology path (EPIET), European Centre for Disease Prevention and Control (ECDC), Solna, Sweden, 3 Centre for International Health Protection, Robert Koch Institute, Berlin, Germany, 4 Direction de la Veille Sanitaire et de la Surveillance Epidémiologique, Ministère de la Santé Publique (DVSSE), Antananarivo, Madagascar, 5 Laboratoire d'Accueil et de Recherche en Santé Publique et en Technologies de l'Information Médicale et de la Communication (LARTIC), Antananarivo, Madagascar, 6 Centre National de Formation et de Recherche en Santé Rurale de Maferinyah, Forécariah, Guinea, 7 Department of Public Health, Gamal Abdel Nasser University of Conakry, Conakry, Guinea, 8 Centre for Biological Threats and Special Pathogens, Robert Koch Institute, Berlin, Germany

☯ These authors contributed equally to this work.
‡ These authors also contributed equally to this work.
* paerischt@posteo.de

## Abstract

### Objectives

To assess training needs of healthcare workers (HCWs) on pulmonary plague (PP) control after the large PP outbreak in Madagascar 2017.

### Methods

In 2018, we conducted a knowledge, attitudes and practices (KAP) survey among HCWs (PP cases and comparison group) in Antananarivo and Toamasina. Proportions were calculated, differences between groups were tested for significance.

### Results

Knowledge levels were similar for HCW PP cases and the HCW comparison group. Among 59 HCW over 90% named the distinctive disease forms of plague (bubonic (93%), pulmonary (98%)), and 72% the causative agent. Washing hands was mentioned as protective measure by 56%, while 93% reported to have always washed hands after performing medical procedures. Only 3.5% reported managed PP cases before the outbreak; 38% reported to have felt confident performing invasive procedures while caring for PP cases at the beginning versus 62% at the end of the

**Data availability statement:** Interested researchers can request the de-identified, anonymized dataset underlying the results presented in this study from raharina4@hotmail.com.

**Funding:** Research for this publication was funded through the Global Health Protection Programme (GHPP) by the Federal Ministry of Health, Germany.

**Competing interests:** The authors have declared that no competing interests exist.

outbreak. HCW who remained uninfected reported more often than PP cases to have worn multiple or single use medical coats ((93% vs. 53%, p = 0.001; 60% vs. 20%, p = 0.028), and less frequently to have paid for chemoprophylaxis out of pocket (11% versus 50%; p = 0.008).

## Conclusion

Despite the good overall knowledge, specific knowledge gaps and the mismatch between knowledge and practice of basic hygiene measures and low confidence in providing care for PP cases after the outbreak indicate a persisting need for training.

---

## Introduction

Plague is considered endemic only in a few regions of the world. Single cases or small outbreaks were reported in the United States of America and in Kyrgyzstan, but the most affected countries according to WHO are the Democratic Republic of Congo, Peru and Madagascar. Over 75% of plague cases worldwide occur in Madagascar [1]. Until 2017, plague cases in Madagascar occurred mostly in rural regions, with a seasonal increase from October to April. Between 1998 and 2016, 88% of the cases reported were classified as bubonic plague (BP) [2]. Despite effective antibiotic treatment being available, prevention and control of the disease remain challenging, leading to its endemic persistence [3–5]. The large outbreak in 2017 was unusual in that it occurred between July and November, produced a large proportion of pulmonary plague (PP) cases, and affected urban areas like Antananarivo and Toamasina. 1878 of the 2412 notified cases (78%) in this time period were classified as PP [6], including 81 HCWs who were diagnosed with and treated for PP according to WHO [7]. Due to potential close contact with infected patients, healthcare workers (HCWs) are at high risk during outbreaks of PP, but only limited literature exists on transmission patterns and risk factors for infection among HCWs, and on their knowledge of the disease and treatment procedures [8–10]. Given the relatively high number of PP cases among HCWs reported during the 2017 Malagasy outbreak (N = 81), we aimed at investigating possible knowledge gaps, unhelpful attitudes and unsafe practices during the outbreak that may have facilitated transmission to HCWs in order to better understand possible causes for infection and develop locally relevant infection prevention and control (IPC) training to be better prepared for future plague outbreaks.

KAP surveys are used to assess knowledge, attitude and practices (KAP) of a selected population regarding a specific topic. They are commonly used in the health sector to gather before-and-after data on public health interventions [11–13]. For emerging diseases in limited resource settings, KAP surveys can help to develop targeted interventions addressing the most relevant problems and improve response measures in case of an outbreak. For example, during the Ebola Virus Disease outbreak in West Africa, KAP surveys contributed to describing and understanding the persisting perceptions and beliefs around the disease in the community [14–16].

There is only limited literature on experiences of HCW on treatment of pulmonary plague in hospital settings [8, 17]. Therefore, we conducted the study to gather more information to support the development of context-adapted strategies for diagnosis, treatment and prevention of nosocomial transmission, with sufficient resources, including trained HCWs available on different levels of care.

## Materials and methods

### Study design

In September 2018, we conducted a cross-sectional survey in four hospitals in the Malagasy capital Antananarivo, and in two hospitals and one mobile team in the Malagasy city Toamasina, which were the most affected urban areas during the outbreak. The KAP survey was conducted in HCWs from the hospital departments where PP cases had arisen among HCWs that were involved in PP treatment, and from mobile teams that were involved in community outreach and body management of patients who died due to PP in the 2017 outbreak.

From 81 HCW plague cases reported by WHO [7], 29 PP cases were selected from the Ministry of Health's national Directorate of Health and Epidemiological Surveillance (DVSSE) database, based on geographical location (Antananarivo and Toamasina) and work in a health care facility at the time of study in 2018. These were contacted and invited to participate in the study (Fig 1).

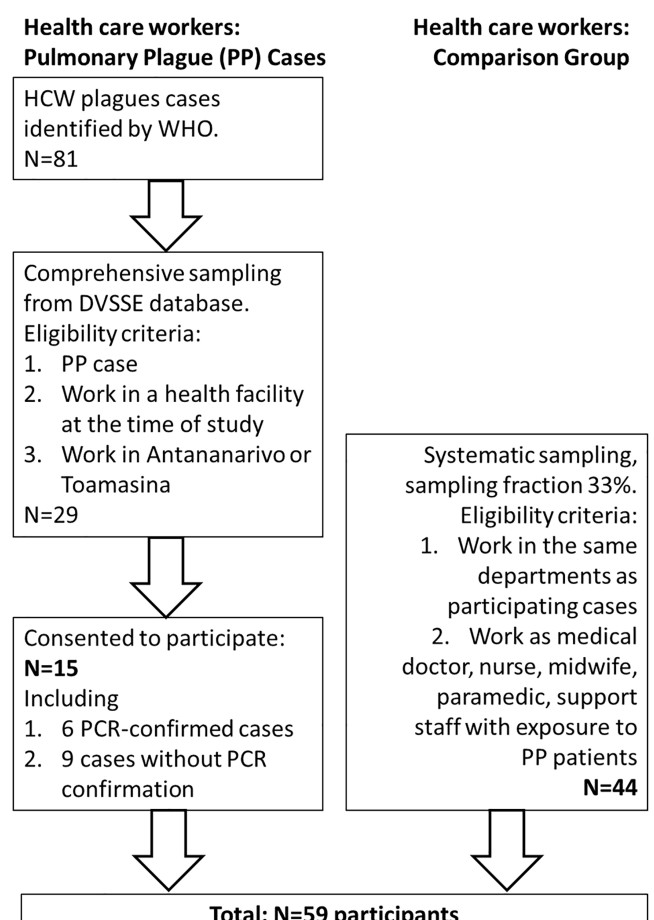

**Fig 1. Sampling scheme.** Sampling scheme of DVSSE – Ministry of Health's national Directorate of Health and Epidemiological Surveillance.

Additional HCWs were recruited as a comparison group from the same departments of hospitals where the participating HCW PP cases worked at the time of the study. Eligible comparison HCWs were defined as medical doctors (including medical interns and students), nursing staff (nurses, midwives, and paramedics), and support staff assumed to have been exposed directly or indirectly to PP patients (cleaners, patient transport staff, morgue staff, guards or mobile community outreach staff). Administrative staff, for whom no patient contact was assumed, were excluded. To avoid selection bias, comparison HCWs were systematically sampled from HCW staff lists of each eligible department. The sample size of N = 44 was determined on feasibility grounds, resulting in a sampling fraction of 33% (Fig 1). If a HCW was selected but absent for a long period of time, for example for holidays, another HCW was randomly sampled from the list.

Upon written informed consent, local interviewers administered a structured questionnaire in French or Malagasy in a designated room to ensure privacy. The participants completed the interview in the language of their choice. Study participants received a token of appreciation for their participation (10 000 Ariary phone credit, equivalent to approx. € 2.50).

## Data collection

The same questionnaire was used on all participants. It consisted of four sections: (1) demographic data (age, sex, work place, profession, and years of working experience), (2) knowledge (causative agent, transmission, infection prevention and control, symptoms, treatment), (3) attitudes towards PP (level of confidence in treatment and protection measures and material). The 4th section consisted of questions on practices during the epidemic (use of existing guidelines, availability of chemoprophylaxis, protection material and treatment). High-risk procedures were defined as suction, intubation, sputum sampling and cleaning of surfaces. Examples for invasive procedures, such as drawing blood and delivery, and examples for non-invasive procedures, such as performing ultrasound or ECG, were provided in the questionnaire. A sheet with photographs of the personal protection equipment was used to assess which items were used. At the end of the interview all HCWs were invited to demonstrate how to wear a surgical mask, which was assessed by a member of the research team.

## Data analysis and definitions

Throughout the epidemic, different case definitions for plague were applied. The DVSSE database we used for recruiting HCW PP cases consisted of cases based on clinical, epidemiological and laboratory criteria.

After double data entry and validation, we used STATA 15 (StataCorp LLC, College Station, Texas, USA) for data management and analysis. A descriptive analysis was performed using ranges, frequency counts (n) and proportions (%).

We looked at differences in self-reported practice during the outbreak between HCW PP cases from the DVSSE database irrespective of the classification as suspected, probable or confirmed case with the comparison group. Due to the changes in the case definition and its unclear effect on the classification of cases in the database, as well as anecdotal evidence of problems with transport of samples and on-site testing with rapid diagnostic tests, we conducted the same comparison on the subset of PCR-confirmed cases versus the corresponding subset of HCWs who worked in the same departments as the PCR positive cases. None of the cases had a positive culture and only one case had a positive serology result recorded in the database.

The threshold for a group having "good knowledge" was defined as ≥75%, while the threshold for a group "lacking knowledge" was defined as <40%.

For better readability, we limit the reporting of digits to two meaningful digits, as in 45%, 4.5% or 0.45%. As a result of this rounding policy, percentages may add up to slightly less or more than 100%. For the comparison of proportions, Fisher's exact test was performed. We applied a significance level of $p < 0.05$, and did not correct for multiple testing since we conducted an explorative study and did not test for any predefined hypothesis.

## Ethics statement

Human participants were included. Written informed consent was obtained from study participants. Ethics clearance was obtained from the Committee of Ethics in Biomedical Research of the Ministry of Health in Madagascar on 30 April 2018 (File No. 048-MSANP/CERBM). A permission to conduct the study within the public health care setting was obtained by the national Ministry of Health of Madagascar. Data protection clearance was granted at Robert Koch Institute, Germany.

## Results

### Characteristics of the study population

From the 29 eligible HCW PP cases from the national DVSSE case database 15 HCW could be traced back and agreed to participate (Fig 1). These HCW were working in seven different institutions in Antananarivo (four hospitals) and Toamasina (two hospitals, one mobile team), where an additional 44 HCWs were recruited as a comparison group, resulting in a total of 59 participants. The remaining 14 HCW PP cases had changed positions or hospitals after the outbreak, were on leave during the study, or refused to participate and could therefore not be included in the study.

From the 59 HCW, 54% were working in Antananarivo and 46% in Toamasina at the time of the survey. More female (63%) than male HCW (37%) were in the sample with an age range of 22–59 years. Most participants were medical doctors including medical interns and students (41%) or nursing staff (41%). Eleven HCWs (19%) were support staff (cleaners, patient transport staff, morgue staff, guards or mobile community outreach staff). The majority of HCWs (78%) reported to have more than five years of working experience, and most confirmed to have received a salary (92%); four volunteers reported to not have received any financial compensation during the outbreak (Table 1).

From the 15 HWCs recorded as PP cases, 6 HCWs had a positive PCR confirming the infection according to the DVSSE database. Five HCWs were from Antananarivo and 10 HCW from Toamasina. The distribution of professional

**Table 1. Demographic characteristics of participating health care workers (HCWs).**

|  | All HCWs (N = 59) | HCW pulmonary plague (PP) cases (N = 15) | HCW comparison group (N = 44) |
|---|---|---|---|
| **City** |  |  |  |
| Antananarivo | 32 (54%) | 5 (33%) | 27 (61%) |
| Toamasina | 27 (46%) | 10 (67%) | 17 (39%) |
| **Gender** |  |  |  |
| male | 22 (37%) | 7 (47%) | 15 (34%) |
| female | 37 (63%) | 8 (53%) | 29 (66%) |
| **Age** range (years) | 22 - 59 | 22 - 58 | 24 - 59 |
| **Profession** |  |  |  |
| Medical Doctors* | 24 (41%) | 7 (47%) | 17 (39%) |
| Nursing staff** | 24 (41%) | 6 (40%) | 18 (41%) |
| Support staff*** | 11 (19%) | 2 (13%) | 9 (21%) |
| **Working experience (years)** |  |  |  |
| < 2 years | 6 (10%) | 2 (13%) | 4 (9.1%) |
| 2-5 years | 10 (17%) | 2 (13%) | 8 (18%) |
| >5 years | 43 (78%) | 11 (73%) | 32 (73%) |
| **Received salary during the outbreak** | 54 (92%) | 12 (80%) | 42 (96%) |

\* including medical interns and students

\*\* Nurses, midwives, and paramedics

\*\*\* Cleaners, patient transport staff, morgue staff, guards and mobile community outreach staff

groups among HCW PP cases and the HCW of the comparison group was similar. The age among the 15 HCW PP cases ranged from 22 to 58 years and among the comparison group from 24 to 59 years (Table 1).

## Knowledge on pulmonary plague

Almost all participants were aware that PP was a contagious disease (98%) to which no immunity is developed (83%, Table 2).

**Table 2. General knowledge among health care workers (HCWs) concerning the disease, prevention and treatment.**

| | All HCWs (N=59)* | | HCW pulmonary plague (PP) cases (N=15) | HCW comparison group (N=44) |
|---|---|---|---|---|
| **To which extent do you agree with the following statement: "Pulmonary plague is a contagious disease"** | | | | **p=1.0 |
| Some agreement | | 1 (1.7%) | 0 (0%) | 1 (2.3%) |
| Total agreement | | 58 (98%) | 15 (100%) | 43 (98%) |
| **To which extent do you agree with the following statement: "Certain people are immune towards PP"** | | | | *p=0.10* |
| Does not know | | 3 (5.1%) | 2 (13%) | 1 (2.3%) |
| Does not agree | | 49 (83%) | 11 (73%) | 38 (86%) |
| Some agreement | | 1 (1.7%) | 1 (6.7%) | 0 (0%) |
| Total agreement | | 6 (10%) | 1 (6.7%) | 5 (11%) |
| **What is the pathogen of PP?** | | | | p=0.65 |
| | | N=58 | N=15 | N=43 |
| Does not know | | 4 (6.9%) | 0 (0%) | 4 (9.3%) |
| Virus | | 11 (19%) | 4 (27%) | 7 (16%) |
| Parasite | | 1 (1.7%) | 0 (0%) | 1 (2.3) |
| Bacteria | | 42 (72%) | 11 (73%) | 31 (72%) |
| **What are different clinical forms of plague? (Please mention the types you know)** | | | | |
| Bubonic | **p=1.0 | 55 (93%) | 14 (93%) | 41 (93%) |
| Pulmonary | p=1.0 | 58 (98%) | 15 (100%) | 43 (98%) |
| Primary or secondary pulmonary | p=1.0 | 4 (6.8%) | 1 (6.7%) | 3 (6.8%) |
| Septicemic | p=0.48 | 12 (20%) | 4 (27%) | 8 (18%) |
| **What are the (main) symptoms of PP? (Please mention the one you know)** | | | | |
| Fever | p=0.16 | 56 (95%) | 13 (87%) | 43 (98%) |
| Dyspnea | p=0.76 | 23 (39%) | 5 (33%) | 18 (41%) |
| Coughing | p=1.0 | 47 (80%) | 12 (80%) | 35 (80%) |
| Chest pain | p=0.11 | 20 (34%) | 8 (53%) | 12 (27%) |
| Expectorations | p=0.48 | 46 (78%) | 13 (87%) | 33 (75%) |
| **Speaking of prevention methods against PP in hospitals, can you tell me which ones you know?** | | | | |
| Isolation | p=0.17 | 44 (75%) | 9 (60%) | 35 (80%) |
| Washing hands | p=0.23 | 33 (56%) | 6 (40%) | 27 (61%) |
| Chemoprophylaxis | p=0.55 | 39 (66%) | 11 (73%) | 28 (64%) |
| Patients wearing a surgical mask | p=0.75 | 40 (68%) | 11 (73%) | 29 (66%) |
| HCWs wearing a mask | p=0.27 | 55 (93%) | 13 (87%) | 42 (95%) |
| Wearing gloves | p=0.48 | 47 (80%) | 11 (73%) | 36 (82%) |
| **Can you name a drug to take in case of chemoprophylaxis?** | | | | |
| Co-trimoxazole | p=0.70 | 49 (83%) | 12 (80%) | 37 (84%) |
| Doxycycline | p=0.22 | 39 (66%) | 12 (80%) | 27 (61%) |
| Ciprofloxacin | p=1.0 | 18 (31%) | 4 (27%) | 14 (32%) |

*N = 59 unless stated otherwise; N < 59 because of missing data.

**P-values derived from Fisher's exact test. **p<0.05 in bold;** *0.10 > p ≥ 0.05 in bold italics*

Most participants knew that the plague is caused by bacteria (72%), but some thought that it is caused by a virus (19%). Few HCWs mentioned the septicemic form of plague (20%) and certain symptoms of PP (chest pain 34%, dyspnea 39%). Most frequently mentioned protective measures were HCWs wearing a mask (93%) and using gloves (80%). Most HCWs knew the following preventive measures: isolating patients (75%), asking patients to wear a mask (68%), taking chemoprophylaxis (66%), and washing hands (56%). When asked about chemoprophylaxis, a majority of HCWs specified co-trimoxazole (83%), followed by doxycycline (66%) and ciprofloxacin (31%). With respect to PP knowledge there were no significant differences between the HCW PP cases and the HCW comparison group.

## Attitudes

The reported confidence in terms of treating patients increased over the course of the outbreak: the proportion of HCWs who reported to have felt totally confident treating PP cases including invasive procedures rose from 38% at the beginning of the outbreak to 62% at its end (Table 3).

A small majority of participants (55%) described their level of knowledge of how to prevent nosocomial transmission of PP as intermediate, and only 10% felt to have expert knowledge at the time of the study. A majority felt that HCW in general were sufficiently informed about the transmission routes of PP (71%). Additionally, 46% considered it to be easy to protect themselves in a hospital setting, and 56% considered it to be easy or very easy to treat PP patients. Only 57% of participants trusted protective equipment fully, while 66% fully trusted chemoprophylaxis. About two thirds reported to have feared contracting PP at work (68%) or outside work (66%). A majority of respondents reported a high workload (79%), and felt that the departments were understaffed during the outbreak (63%). HCW PP cases trusted chemoprophylaxis significantly less than the HCW comparison group (47% vs. 73%, p = 0.041). Fewer HCW PP cases than comparison HCWs found it easy to protect themselves from PP transmission in a hospital (20% vs. 55%) and thought HCWs in general to be sufficiently informed about the transmission of PP (53% vs. 77%), while more HCW PP case than comparison HCWs felt that they were understaffed during the epidemic (87% vs. 55%), but these difference failed to gain statistical significance (0.10 > p ≥ 0.05).

## Practices

The most common clinical practices were clinical examinations (42%), invasive procedures (39%) and local disinfection (39%). Among high-risk procedures the most common were surface cleaning in treatment centers (37%), followed by sputum sampling (32%) and intubation (25%, Table 4).

Most HCWs (95%) reported to have followed guidelines during the outbreak. A small majority (54%) of respondents stated that the patients always wore surgical masks during medical care. In regards to their own practice, 46% of the participants responded that they always wore a surgical mask, and 46% to have always worn an FFP2 mask while caring for patients. A large majority of the participants reported to have always worn gloves (86%) and to have always used a multiple use coat (80%). Most HCWs (93%) reported to have always washed their hands after high-risk procedures. Patients with PP were almost always isolated according to participants (97%). The use of chemoprophylaxis during the epidemic was common among the participants (91%). The most common antibiotic used was co-trimoxazole (59%). A minority of participants who reported taking chemoprophylaxis paid out of pocket for it (20%). Almost a quarter of the participants paid themselves for protective material (23%). 49% of the HCWs reported to have taken care of a PP case that they knew personally.

Only 2 HCWs (3.5%) had previous experience in working with PP cases before the 2017 epidemic. A minority of the HCWs reported to have had PP classes or training before the epidemic in 2017 (21%). During the epidemic 50% received training. After the epidemic, 21% attended training on health care provision for PP patients. In total, 26% of HCW had not received any PP training. Only about half of the participants (48%) could correctly demonstrate the use of a surgical mask.

A significantly lower proportion of the cases compared to the comparison group reported to always have worn the multiple use medical coats (53% vs. 93%, p = 0.001) or a single use medical coat (20% vs. 60%; p = 0.028). A significantly

**Table 3. Attitudes towards pulmonary plague (PP) among health care workers (HCWs).**

| | All HCWs (N=59)* | HCW PP cases (N=15) | HCW comparison group (N=44) |
|---|---|---|---|
| **Indicate the extent to which you agree with the following statements:** | | | |
| "I was confident in performing the high-risk procedures <u>at the beginning</u> of the 2017 epidemic" | | | **p=0.46 |
| | N=50 | N=12 | N=38 |
| Total agreement | 19 (38%) | 5 (42%) | 14 (37%) |
| Some agreement | 7 (14%) | 0 (0%) | 7 (18%) |
| No agreement | 23 (46%) | 7 (58%) | 16 (42%) |
| I don't know | 1 (2.0%) | 0 (0%) | 1 (2.6%) |
| "I was confident in performing the high-risk procedures <u>towards the end</u> of the 2017 epidemic" | | | p=0.41 |
| | N=50 | N=12 | N=38 |
| Total agreement | 31 (62%) | 6 (50%) | 25 (66%) |
| Some agreement | 9 (18%) | 2 (17%) | 7 (18%) |
| No agreement | 10 (20%) | 4 (33%) | 6 (16%) |
| **How would you describe your knowledge of preventing nosocomial/hospital PP transmission?** | | | p=0.50 |
| | N=58 | N=15 | N=43 |
| Expert knowledge | 6 (10%) | 1 (6.7%) | 5 (12%) |
| Intermediate knowledge | 32 (55%) | 9 (60%) | 23 (53%) |
| Basic knowledge | 15 (26%) | 3 (20%) | 12 (28%) |
| No knowledge | 3 (5.2%) | 2 (13%) | 1 (2.3%) |
| I don't know | 2 (3.5%) | 0 (0%) | 2 (4.7%) |
| **How easy is it to protect yourself from transmission of PP in a hospital setting?** | | | *p=0.067* |
| Very easy | 0 (0%) | 0 (0%) | 0 (0%) |
| Easy | 27 (46%) | 3 (20%) | 24 (55%) |
| Average | 11 (19%) | 3 (20%) | 8 (18%) |
| Difficult | 14 (24%) | 6 (40%) | 8 (18%) |
| Very difficult | 7 (12%) | 3 (20%) | 4 (9.1%) |
| **How easy is it to treat patients with pneumonic plague in a hospital setting?** | | | p=0.29 |
| Very easy | 4 (6.8%) | 3 (20%) | 1 (2.3%) |
| Easy | 29 (49%) | 6 (40%) | 23 (52%) |
| Average | 6 (10%) | 1 (6.7%) | 5 (11%) |
| Difficult | 17 (29%) | 5 (33%) | 12 (27%) |
| Very difficult | 1 (1.7%) | 0 (0%) | 1 (2.3%) |
| I don't know | 2 (3.4%) | 0 (0%) | 2 (4.6%) |
| **Indicate the extent to which you agree with the following statement:** | | | |
| "I trust the personal protective equipment." | | | p=0.13 |
| | N=58 | N=14 | N=44 |
| Total agreement | 33 (57%) | 8 (57%) | 25 (57%) |
| Some agreement | 17 (29%) | 2 (14%) | 15 (34%) |
| No agreement | 8 (14%) | 4 (29%) | 4 (9.1%) |
| "I trust the drugs for chemoprophylaxis." | | | **p=0.041** |
| Total agreement | 39 (66%) | 7 (47%) | 32 (73%) |
| Some agreement | 18 (31%) | 6 (40%) | 12 (27%) |
| No agreement | 2 (3.4%) | 2 (13%) | 0 (0%) |
| "I was afraid of getting infected with PP <u>at work (in the hospital)</u> during the epidemic." | | | p=0.57 |
| Total agreement | 40 (68%) | 10 (67%) | 30 (68%) |
| Some agreement | 4 (6.8%) | 0 (0%) | 4 (9.1%) |

*(Continued)*

**Table 3.** (Continued)

| | All HCWs (N=59)* | HCW PP cases (N=15) | HCW comparison group (N=44) |
|---|---|---|---|
| No agreement | 14 (24%) | 5 (33%) | 9 (20%) |
| I don't know | 1 (1.7%) | 0 (0%) | 1 (2.3%) |
| "I was afraid of contracting pneumonic plague <u>outside work</u> during the epidemic." | | | p=0.44 |
| Total agreement | 39 (66%) | 9 (60%) | 30 (68%) |
| Some agreement | 3 (5.1%) | 0 (0%) | 3 (6.8%) |
| No agreement | 17 (29%) | 6 (40%) | 11 (25%) |
| "During the epidemic the workload was too high." | | | p=0.85 |
| | N=57 | N=15 | N=42 |
| Total agreement | 45 (79%) | 13 (87%) | 32 (76%) |
| Some agreement | 4 (7.0%) | 1 (6.7%) | 3 (7.1%) |
| No agreement | 8 (14%) | 1 (6.7%) | 7 (17%) |
| "During the epidemic we were understaffed." | | | *p=0.088* |
| | N=57 | N=15 | N=42 |
| Total agreement | 36 (63%) | 13 (87%) | 23 (55%) |
| Some agreement | 3 (5.3%) | 0 (0%) | 3 (7.1%) |
| No agreement | 18 (32%) | 2 (13%) | 16 (38%) |
| **Are health workers (in general) sufficiently informed about the transmission of PP?** | | | *p=0.051* |
| Yes | 42 (71%) | 8 (53%) | 34 (77%) |
| No | 14 (24%) | 7 (47%) | 7 (16%) |
| I don't know | 3 (5.1%) | 0 (0%) | 3 (6.8%) |

*N=59 unless stated otherwise; N<59 because of missing data.

**P-values derived from Fisher's exact test. **p<0.05 in bold;** *0.10>p≥0.05 in bold italics*

higher proportion of the cases reported to have paid for chemoprophylaxis out of their own pockets (50% vs. 11%; p = 0.008). Fewer HCW PP cases than comparison HCWs reported to have followed guidelines or protocols for managing PP patients or preventing PP transmission (87% vs. 98%), but this difference was not significant (0.10 > p ≥ 0.05). Other practices did not differ between HCW PP cases and the HCW comparison group.

We conducted a subgroup analysis of six PCR confirmed cases and their corresponding comparison group of 26 HCWs. We found a significant difference for reported usage of multiple use medical coats (50% versus 100%; p = 0.004; data not shown). A significantly higher proportion of PCR confirmed cases than comparison HCW received training after the outbreak (67% versus 20%; p = 0.043).

## Discussion

This study provides information on knowledge, attitudes and practices during the plague outbreak in 2017 and highlights the gaps in IPC training and implementation of IPC measures among HCW in urban areas in Madagascar. We found basic PP knowledge to be overall good. Prevention measures like wearing masks and gloves and isolating patients were well known (≥75% of participants). Some theoretical knowledge, however, was lacking, such as: knowing about the pulmonary and bubonic but not the septicemic form of plague, knowing the unspecific symptom "fever" but not common symptoms "dyspnea" or "chest pain" (<40% of participants). HCWs felt confident about their abilities to treat patients and perceived a learning effect by working during the outbreak. Nonetheless, the results indicate room for improvement, and there remain training needs concerning disease characteristics, necessary IPC measures and their purpose, and the correct use of personal protective equipment.

**Table 4. Self-reported practices of health care workers (HCWs) during the plague epidemic in 2017 (ctd.).**

| | All HCWs | HCW PP cases | HCW comparison group |
|---|---|---|---|
| **Clinical practice** | N = 59 | N = 15 | N = 44 |
| "What were you main activities in the hospital during the epidemic"? | | | |
| Clinical examination | | | p = 0.77 |
| Yes | 25 (42%) | 7 (47%) | 18 (41%) |
| No | 34 (58%) | 8 (53%) | 26 (59%) |
| Non-invasive care (e.g., ultrasound, ECG) | | | p = 1.0 |
| Yes | 8 (14%) | 2 (13%) | 6 (14%) |
| No | 51 (86%) | 13 (87%) | 38 (86%) |
| Invasive care (e.g., blood sampling, delivery) | | | p = 0.55 |
| Yes | 23 (39%) | 7 (47%) | 16 (36%) |
| No | 36 (61%) | 8 (53%) | 28 (64%) |
| Local disinfection | | | p = 1.0 |
| Yes | 23 (39%) | 6 (40%) | 17 (39%) |
| No | 36 (61%) | 9 (60%) | 27 (61%) |
| Transport of sick patients | | | p = 1.0 |
| Yes | 10 (17%) | 2 (13%) | 8 (18%) |
| No | 49 (83%) | 13 (87%) | 36 (82%) |
| Dead body management | | | p = 0.68 |
| Yes | 9 (15%) | 3 (20%) | 6 (14%) |
| No | 50 (85%) | 12 (80%) | 38 (86%) |
| "Which high-risk procedures did you execute?" | | | |
| Suction | | | p = 1.0 |
| Yes | 1 (1.7%) | 0 (0%) | 1 (2.3%) |
| No | 58 (98%) | 15 (100%) | 43 (98%) |
| Intubation | | | p = 0.31 |
| Yes | 15 (25%) | 2 (13%) | 13 (30%) |
| No | 44 (75%) | 13 (87%) | 31 (70%) |
| Sputum sampling | | | p = 0.34 |
| Yes | 19 (32%) | 3 (20%) | 16 (36%) |
| No | 40 (68%) | 12 (80%) | 28 (64%) |
| Cleaning of surfaces in a treatment center | | | p = 1.0 |
| Yes | 22 (37%) | 6 (40%) | 16 (36%) |
| No | 37 (62%) | 9 (60%) | 28 (63%) |
| **Experience in care for PP cases prior to the outbreak, and participation in PP training** | | | |
| | N = 58 | N = 15 | N = 43 |
| Did you ever manage cases of PP before the 2017 outbreak? | | | p = 1.0 |
| Yes | 2 (3.5%) | 0 (0%) | 2 (4.6%) |
| No | 55 (95%) | 15 (100%) | 40 (93%) |
| I don't know | 1 (1.7%) | 0 (0%) | 1 (2.3%) |
| Did you participate in PP courses/trainings <u>before</u> the outbreak? | | | p = 0.16 |
| Yes | 12 (21%) | 1 (6.7%) | 11 (26%) |
| No | 46 (79%) | 14 (93%) | 32 (74%) |
| Did you participate in PP courses/trainings <u>during</u> the outbreak? | | | p = 1.0 |
| Yes | 29 (50%) | 8 (53%) | 21 (49%) |
| No | 27 (47%) | 7 (47%) | 20 (47%) |
| I don't know | 2 (3.5%) | 0 (0%) | 2 (4.7%) |

*(Continued)*

**Table 4.** (Continued)

| | All HCWs | HCW PP cases | HCW comparison group |
|---|---|---|---|
| Did you participate in PP courses/trainings <u>after</u> the outbreak? | | | p = 0.31 |
| Yes | 12 (21%) | 5 (33%) | 7 (16%) |
| No | 44 (76%) | 10 (67%) | 34 (79%) |
| I don't know | 2 (3.5%) | 0 (0%) | 2 (4.7%) |
| Did you participate in PP courses/trainings <u>at any point in time</u>? | | | p = 1.0 |
| | N = 57 | N = 15 | N = 42 |
| Yes | 42 (74%) | 11 (73%) | 31 (74%) |
| No | 15 (26%) | 4 (27%) | 11 (26%) |
| **Demonstration of use of a surgical masque** | | | p = 0.37 |
| | N = 58 | N = 15 | N = 43 |
| Correct | 28 (48%) | 9 (60%) | 19 (44%) |
| Wrong | 30 (52%) | 6 (40%) | 24 (56%) |

*N = 59 unless stated otherwise; N < 59 because of missing data.

**P-values derived from Fisher's exact test. **p < 0.05 in bold;** *0.10 > p ≥ 0.05 in bold italics*

There were several discrepancies between knowledge and reported practice. First of all, transmission prevention measures like hand washing after providing care for a PP case were mentioned by only half of the HCWs, but when asked about hand hygiene directly, virtually all reported to have practiced it during the epidemic after performing a high-risk procedure. This gap between knowledge and reported practice suggests that HCWs' reported practices may be overstated compared to actions performed in reality. Alternatively, they may have practiced hand hygiene more often than naming it as preventive measure, suggesting that it may not have been fully understood why certain procedures are important while caring for PP cases. Conversely, a higher proportion of HCWs knew that suspected cases should wear a mask (68%), but a lower proportion reported that this was always done in practice (54%). Also, wearing a mask by HCWs themselves was mentioned by 93% as a protective measure, while only half of them reported to have always worn a surgical or FFP2 mask (46% and 46%, respectively) while caring for patients. In addition, only 48% could demonstrate how to correctly wear a surgical mask. This gap between knowledge and practice in relation to patients may indicate a lack of implementation of, or compliance with, guidelines despite better knowledge, or insufficient availability of protective equipment such as masks. The gap may also be due to the fact that only few HCWs had training before or after the epidemic on handling PP patients. Further training, availability of biosafety guidelines and access to sufficient PPE are likely to increase staff safety and confidence [18–20].

A fifth of the HCW participants report to have paid for chemoprophylaxis themselves. This is worrying, since chemoprophylaxis should be provided to exposed HCWs for free. Whether chemoprophylaxis was unavailable to these HCWs, or whether their initiative reflected their fears, could not be distinguished by our study. However, the difference between HCW PP cases and HCW comparison group was particularly striking in this respect: an almost five times higher proportion of cases reported out-of-pocket payment for chemoprophylaxis than HCWs from the comparison group. This may suggest that cases have experienced access barriers to chemoprophylaxis and were, therefore, less well protected against infection. We therefore recommend ensuring equal access to chemoprophylaxis for all exposed HCWs.

The high-risk of contracting PP from working in the hospital or mobile team in the community was perceived as similar to contracting PP from outside the work context. The analysis of 2414 plague cases, 78% PP cases and 16% BP cases, between August and November 2017 did not reveal any recorded nosocomial transmission [6]. Our study population seemed to have overall good knowledge of various aspects of the disease, including its clinical manifestations, and of

adequate infection prevention practices. While only a small proportion of the HCWs had received training during and after the epidemic, and a considerable proportion had not received any training at all, the hands-on experience may have led to reported practice being better than theoretical knowledge, and to confidence to treat patients with PP to grow in the course of the epidemic. Worryingly, only half of the HCW (48%) could demonstrate the correct use of a surgical mask, which suggests a persisting need for hands-on training.

The case definition changed over the evolution of the outbreak, which may have contributed to misclassification if HCW were not trained accordingly. Lack of knowledge of case definitions, possibly in combination with lack of knowledge and training on PP, may have contributed to possible misclassification of HCW as PP cases. A sensitive case definition ensured the capture of all suspected cases of PP, which is desirable due to the fast evolution and severity of disease. However, with limited capacity for adequate testing despite having samples from the majority of cases, only 2% of the clinically suspected PP cases in 2017 could be laboratory confirmed [6]. Consequently, the number of PP cases may have been overestimated.

This study has various limitations. We interviewed the HCWs in their working environment, and although we ensured privacy for the interview of each HCW to minimize courtesy bias, they may have been reluctant to voice criticism or report mismanagement for fear of retributions. Our study was carried out one year after the outbreak, and we used photographic prompts to minimize recall bias. Still, our study cannot distinguish with confidence between knowledge and attitudes acquired during vs. after the outbreak, and residual courtesy and recall bias may have embellished the answers on practices during the outbreak. Changes of case definition and availability and reliability of diagnostic tests over the course of the epidemic may have influenced the inclusion of PP cases in the DVSSE database. We could only include those HCW PP cases that remained in the same unit after the outbreak and were available at the time of the study; these may not be fully representative of all HCW PP cases.

## Conclusions

This study shows that one year after the 2017 plague epidemic, HCWs in urban areas in Madagascar had an overall good level of knowledge, but only half reported to feel confident treating PP cases despite the hands-on experience during the outbreak. Our results indicate that there may be a persistent need among HCWs for training in IPC methods, such as hand hygiene, PPE use, and adequate isolation, to protect both HCW and patients. Despite awareness of existing guidelines, standard operating procedures for PP-specific PPE-use and management of PP cases tailored to the Malagasy setting may contribute to being better prepared for future large-scale outbreaks.

## Supporting information

**S1 File. Questionnaire "Inclusivity in global research".**
(DOCX)

## Acknowledgments

We would like to thank the health care workers who dedicated, despite their heavy workload, significant time to the study team and to make this survey possible. Furthermore, we would like to extend our gratitude to the DVSSE, LARTIC and Pasteur Institute for their invaluable contribution to this study.

## Author contributions

**Conceptualization:** Inessa Markus, Lynn Meurs, Vaoary Razafimbia, Andrianasolo Radonirina, Rapelerano Rabenja Fahafahantsoa, Alexandre Delamou, Matthias Borchert, Thomas Paerisch.
**Data curation:** Inessa Markus, Lynn Meurs, Rebekah Wood, Vaoary Razafimbia, Delphin Kolie, Thomas Paerisch.

**Formal analysis:** Inessa Markus, Lynn Meurs, Rebekah Wood.

**Funding acquisition:** Matthias Borchert, Thomas Paerisch.

**Investigation:** Inessa Markus, Lynn Meurs.

**Methodology:** Inessa Markus, Lynn Meurs, Andrianasolo Radonirina, Matthias Borchert.

**Project administration:** Thomas Paerisch.

**Software:** Rebekah Wood.

**Supervision:** Rapelerano Rabenja Fahafahantsoa, Jan Walter, Matthias Borchert, Thomas Paerisch.

**Writing – original draft:** Inessa Markus, Lynn Meurs, Rebekah Wood, Vaoary Razafimbia, Andrianasolo Radonirina, Delphin Kolie, Jan Walter, Matthias Borchert.

**Writing – review & editing:** Inessa Markus, Lynn Meurs, Vaoary Razafimbia, Andrianasolo Radonirina, Delphin Kolie, Alexandre Delamou, Jan Walter, Matthias Borchert, Thomas Paerisch.

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
