## [Decision Letter · Decision Letter 0]

9 Apr 2025

PONE-D-25-02320Knowledge, Attitude and Practices among Healthcare Workers towards Pulmonary Plague Infection following an Outbreak in Madagascar, 2017PLOS ONE

Dear Dr. Paerisch,

Thank you for submitting your manuscript to PLOS ONE. After careful consideration, we feel that it has merit but does not fully meet PLOS ONE’s publication criteria as it currently stands. Therefore, we invite you to submit a revised version of the manuscript that addresses the points raised during the review process.

We look forward to receiving your revised manuscript.

Kind regards,

Beza Ramasindrazana

Academic Editor

PLOS ONE

**Journal Requirements:**

1. When submitting your revision, we need you to address these additional requirements. Please ensure that your manuscript meets PLOS ONE's style requirements, including those for file naming. The PLOS ONE style templates can be found at https://journals.plos.org/plosone/s/file?id=wjVg/PLOSOne_formatting_sample_main_body.pdf and https://journals.plos.org/plosone/s/file?id=ba62/PLOSOne_formatting_sample_title_authors_affiliations.pdf 2. Please include a complete copy of PLOS’ questionnaire on inclusivity in global research in your revised manuscript. Our policy for research in this area aims to improve transparency in the reporting of research performed outside of researchers’ own country or community. The policy applies to researchers who have travelled to a different country to conduct research, research with Indigenous populations or their lands, and research on cultural artefacts. The questionnaire can also be requested at the journal’s discretion for any other submissions, even if these conditions are not met. Please find more information on the policy and a link to download a blank copy of the questionnaire here: https://journals.plos.org/plosone/s/best-practices-in-research-reporting. Please upload a completed version of your questionnaire as Supporting Information when you resubmit your manuscript. 3. We note that the grant information you provided in the ‘Funding Information’ and ‘Financial Disclosure’ sections do not match.  When you resubmit, please ensure that you provide the correct grant numbers for the awards you received for your study in the ‘Funding Information’ section. 4. Thank you for stating the following financial disclosure: Research for this publication was funded through the Global Health Protection Programme (GHPP) by the Federal Ministry of Health, Germany  Please state what role the funders took in the study.  If the funders had no role, please state: "The funders had no role in study design, data collection and analysis, decision to publish, or preparation of the manuscript." If this statement is not correct you must amend it as needed. Please include this amended Role of Funder statement in your cover letter; we will change the online submission form on your behalf. 5. In the online submission form, you indicated that The data underlying the results presented in the study are available from the corresponding author. All PLOS journals now require all data underlying the findings described in their manuscript to be freely available to other researchers, either a. In a public repository, b. Within the manuscript itself, or c. Uploaded as supplementary information.This policy applies to all data except where public deposition would breach compliance with the protocol approved by your research ethics board. If your data cannot be made publicly available for ethical or legal reasons (e.g., public availability would compromise patient privacy), please explain your reasons on resubmission and your exemption request will be escalated for approval. 6. Please include your full ethics statement in the ‘Methods’ section of your manuscript file. In your statement, please include the full name of the IRB or ethics committee who approved or waived your study, as well as whether or not you obtained informed written or verbal consent. If consent was waived for your study, please include this information in your statement as well.

Reviewers' comments:

Reviewer's Responses to Questions

**Comments to the Author**

1. Is the manuscript technically sound, and do the data support the conclusions?

Reviewer #1: No

Reviewer #2: Yes

2. Has the statistical analysis been performed appropriately and rigorously? 

Reviewer #1: No

Reviewer #2: Yes

3. Have the authors made all data underlying the findings in their manuscript fully available?

Reviewer #1: No

Reviewer #2: Yes

4. Is the manuscript presented in an intelligible fashion and written in standard English?

Reviewer #1: Yes

Reviewer #2: No

5. Review Comments to the Author

**Reviewer #1:**  Knowledge, Attitude and Practices among Healthcare Workers towards Pulmonary

Plague Infection following an Outbreak in Madagascar, 2017

The study aimed to assess training needs of healthcare workers (HCWs) on pulmonary plague (PP) control after the Madagascar 2017 large PP outbreak. The authors in 2018, conducted a knowledge, attitudes and practices (KAP) survey among HCWs (PP cases and comparison group) in Antananarivo and Toamasina.

General comments: The study is described as comparative between PP cases and the comparison group, but the rationale for the sample size selection needs further clarification. The study includes only 15 PP cases and 44 comparison group participants, which may limit the representativeness of the results for the entire population of PP cases. Therefore, this study should be presented as a pilot study, due to the small sample size. I therefore suggest the following title : "Knowledge, Attitude and Practices among Healthcare Workers towards Pulmonary Plague Infection following an Outbreak in Madagascar, 2017 : A pilot study "

The demographic data are presented for PP cases and the comparison group, but not all results are compared between PP cases and the comparison group. The methodology requires further clarification to ensure the rigor and reproducibility of the study.

The percentages shown in the tables are often inconsistent, sometimes totaling less than or exceeding 100%. The authors note that the study shows that one year after the 2017 plague epidemic, HCWs in urban areas in Madagascar had a good level of knowledge. However, they do not specify the criteria used to determine a "good level of knowledge."

The authors should specify whether the data from this study are available, in a Data Availability section.

To improve the quality of the manuscript, we encourage the authors to incorporate the following suggestions.

ABSTRACT:

The abstract accurately reflects the objectives, findings, and provides a clear overview of the background, and key results. The results could be strengthened by emphasizing the comparison of plague knowledge between PP cases and the comparison group.

INTRODUCTION

The paper provides adequate background on plague, KAP (Knowledge, Attitudes, and Practices) surveys, and the 2017 outbreak in Madagascar.

Line 51: "Despite the good overall knowledge" � What are the criteria used to define "good overall knowledge"? This should be clarified in the methodology section.

Line 72: "Relatively high number of PP cases among HCWs reported during the 2017 Malagasy outbreak." Please specify the number or prevalence.

Line 86: "Only limited literature on experiences of HCWs on treatment of pulmonary plague in hospital." Please cite some examples if they exist.

MATERIALS AND METHODS

The methods are appropriate for the study’s objectives. The data collection and analysis techniques are generally well-described. However, I suggest adding a detailed explanation of the sample size calculation to enhance reproducibility.

Study design:

Line 93: "We conducted a cross-sectional survey in hospitals..." � How many hospitals were included in the study ?

Line 98: "Recruitment of study participants was conducted between 13 and 27 August 2018." � There was a one-year gap between the end of the 2017 epidemic and the surveys. This is therefore a retrospective study. Please clarify how recall bias was controlled during data collection.

Line 108: "Comparison HCWs were systematically sampled from staff lists to recruit 33% of each department’s HCWs." � Why was this 33% proportion chosen? Is it supported by literature review ? If so, please provide a reference.

Data collection:

Line 116: "The first three sections applied to all participants..." Further, in Line 120, "HCWs who had worked in the hospital or a mobile unit during the 2017 epidemic also completed the 4th section, which included questions on practices." � Please specify how many participants completed each section.

Lines 127-129: "The 5th section was only completed by HCWs who were notified as a PP case during the outbreak and contained questions about the disease, such as symptoms, onset of symptoms, and treatment." � However, questions about symptoms and treatment are addressed in Table 2 for all 59 participants. Does this imply that additional analyses were conducted but are not presented in this article?

RESULTS

The results are clearly presented but require a reorganization in how they are presented. The description of the tables should be provided in full, and the corresponding table should be inserted after the text. Currently, the authors partially describe the table, insert it, and then continue with the description. This approach disrupts the flow and should be revised to enhance clarity and readability.

Characteristics of the study population

Line 155: "These HCWs were working in seven different institutions..." � Please specify these institutions.

In Table 1, for the profession "MD" Please write out the full term (e.g., "Medical Doctor").

Lines 172-173: "The distribution of professional groups among HCW PP cases and the HCW of the comparison group was comparable." � By "comparable," do you mean « similar »?

Line 178, Table 2: The study is presented as a comparative analysis between PP cases and the comparison group. � So, the authors should provide a comparative table between PP cases and the comparison group. If not, please specify in the methodology section, which analyses involve comparisons between PP cases and the comparison group and justify why comparisons are not made for all analyses.

Verify/Recalculate the proportions in Table 2. For example, "Some agreement" = 1 (1.7%) and "Total agreement" = 58 (98%), which totals 99.7% instead of 100%.

For the question, "To what extent do you agree with the following statement: 'Certain people are immune towards PP'," the total is 101.3% instead of 100%.

Tables 3 and 4: � The authors should provide a comparative table between PP cases and the comparison group. Also, verify/recalculate the proportions in these tables.

Table 5: Verify/recalculate the proportions in the table. Many proportions are either above or below 100%.

Lines 237-238: "After the epidemic, 21% attended training on health care provision for PP patients (Table 4)." � Elements of interpretation for Table 4 should be included before Table 4, not after Table 5.

DISCUSSION

Line 257: "We found basic PP knowledge to be good" �Specify in the methodology the criteria used to define good or poor knowledge in this study.

**Reviewer #2:**  This manuscript addresses an important and underexplored topic regarding the knowledge, attitudes, and practices of healthcare workers (HCWs) during a pulmonary plague outbreak in Madagascar. The study design is valuable and the inclusion of diverse HCW roles—from doctors to cleaning staff—adds depth. I appreciate that the authors addressed comments from a previous version.

However, there are still sections that would benefit from clearer articulation, particularly around the description of the study groups and case definitions. The paragraph explaining the comparison of PCR-confirmed cases versus other HCWs, for instance, is essential but currently difficult to follow. A diagram illustrating the sampling process and case categorization would greatly improve clarity. Additionally, some aspects of the questionnaire distribution across diverse HCW roles (e.g., clinical vs. non-clinical staff) could be justified further, especially if the same questions were applied across the board.

I also suggest expanding the interpretation of findings regarding access to chemoprophylaxis. For instance, HCW PP cases may have sought medication on their own due to fear or perceived urgency, not only due to systemic barriers. Highlighting such nuances would strengthen the discussion.

Overall, this study makes an important contribution, but some refinement in structure and clarity would help convey the message more effectively.

6. PLOS authors have the option to publish the peer review history of their article (what does this mean? ). If published, this will include your full peer review and any attached files.

**Do you want your identity to be public for this peer review?** For information about this choice, including consent withdrawal, please see our Privacy Policy .

Reviewer #1: No

Reviewer #2: No

---

## [Author Response · Author response to Decision Letter 1]

14 Oct 2025

Dear reviewers,

I hope this message finds you well.

My co-author team and I have responded to your concerns in a "response to reviewers" letter as well as incorporating changes in the manuscript. Due to challenges in reaching out to the different co-authors, this took a bit of time, - I would like to apologize for the delay. Please do not hesitate to contact me anytime for further questions. Best regards, Thomas Paerisch

---

## [Editor Report · Decision Letter 1]

29 Oct 2025

Knowledge, attitude and practices among Healthcare Workers towards Pulmonary Plague Infection following an outbreak in Madagascar, 2017: A pilot study

PONE-D-25-02320R1

Dear Dr. Thomas Paerisch,

We’re pleased to inform you that your manuscript has been judged scientifically suitable for publication and will be formally accepted for publication once it meets all outstanding technical requirements.

Kind regards,

Beza Ramasindrazana

Academic Editor

PLOS ONE

Additional Editor Comments (optional):

Thank you very much for the responses you provided following reviewers' comments and suggestions. I think that your manuscript is now suitable for publication in PlosOne.

---

## [Editor Report · Acceptance letter]

PONE-D-25-02320R1

PLOS ONE

Dear Dr. Paerisch,

I'm pleased to inform you that your manuscript has been deemed suitable for publication in PLOS ONE. Congratulations! Your manuscript is now being handed over to our production team.

Kind regards,

on behalf of

Dr. Beza Ramasindrazana

Academic Editor

PLOS ONE